# Endocrine Disorders in Nephrotic Syndrome—A Comprehensive Review

**DOI:** 10.3390/biomedicines12081860

**Published:** 2024-08-15

**Authors:** Maja Mizdrak, Bozo Smajic, Ivan Mizdrak, Tina Ticinovic Kurir, Marko Kumric, Ivan Paladin, Darko Batistic, Josko Bozic

**Affiliations:** 1Department of Internal Medicine, University Hospital of Split, 21000 Split, Croatiatticinov@mefst.hr (T.T.K.); 2Department of Pathophysiology, University of Split School of Medicine, 21000 Split, Croatia; marko.kumric@mefst.hr (M.K.); josko.bozic@mefst.hr (J.B.); 3Department of Otorhinolaryngology, Head and Neck Surgery, University of Split School of Medicine, 21000 Split, Croatia; imizdrak@kbsplit.hr (I.M.); ivan.paladin@gmail.com (I.P.); 4Laboratory for Cardiometabolic Research, University of Split School of Medicine, 21000 Split, Croatia; 5Department of Ophthalmology, University Hospital of Split, 21000 Split, Croatia; batisticdarko@gmail.com

**Keywords:** endocrine gland, nephrotic syndrome, pathophysiology, thyroid gland, gonadal axis, hypophysis, adrenal gland, pancreas, parathyroid gland

## Abstract

Nephrotic syndrome is a clinical syndrome characterized by massive proteinuria, called nephrotic range proteinuria (over 3.5 g per day in adults or 40 mg/m^2^ per hour in children), hypoalbuminemia, oncotic edema, and hyperlipidemia, with an increasing incidence over several years. Nephrotic syndrome carries severe morbidity and mortality risk. The main pathophysiological event in nephrotic syndrome is increased glomerular permeability due to immunological, paraneoplastic, genetic, or infective triggers. Because of the marked increase in the glomerular permeability to macromolecules and the associated urinary loss of albumins and hormone-binding proteins, many metabolic and endocrine abnormalities are present. Some of them are well known, such as overt or subclinical hypothyroidism, growth hormone depletion, lack of testosterone, vitamin D, and calcium deficiency. The exact prevalence of these disorders is unknown because of the complexity of the human endocrine system and the differences in their prevalence. This review aims to comprehensively analyze all potential endocrine and hormonal complications of nephrotic syndrome and, vice versa, possible kidney complications of endocrine diseases that might remain unrecognized in everyday clinical practice.

## 1. Introduction

Nephrotic syndrome (NS) is a clinical syndrome characterized by massive proteinuria (over 3.5 g per day in adults or 40 mg/m^2^ per hour in children), hypoalbuminemia, edema, and hyperlipidemia. It has an incidence of 1.15–16.9 cases/100,000 children and 2.7–4 cases/100,000 adults per year, varying by region and ethnicity [1,2,3,4]. The main pathophysiological event in NS is the injury of podocytes, as main parts of the glomerular filtration barrier, and consecutive increased glomerular permeability for proteins, which allows passage of albumins and other proteins into the urine [5]. Proteinuria then causes a cascade of complications, such as fluid accumulation, kidney inflammation, blood pressure dysregulation, hypercoagulable state, infections, immunity disruptions, the increased risk of atherosclerosis, and cardiovascular complications. NS can be caused by different immune triggers, infections, systemic circulating factors, malignant and metabolic diseases, as well as inherited structural abnormalities [6]. Common pathohistological findings are membranous nephropathy (MN), minimal change disease, diabetic nephropathy, and focal segmental glomerulosclerosis, but the definitive diagnosis is found by kidney biopsy [7]. It is associated with high morbidity despite notable advances in treatment strategies and novel medication development. Over time, one-year mortality of NS was stable, accounting for 13–16% [4]. NS eventually leads to severe renal complications, including a progressive decline in kidney function, glomerulosclerosis, tubular atrophy, and interstitial fibrosis [8]. As NS is characterized by urinary loss of hormone-binding proteins, many metabolic disorders and endocrine abnormalities are common. The exact percentage of endocrine disorders in NS is not known due to the complexity of the endocrine system in humans. According to the previous knowledge, we assume that there is a bidirectional connection between endocrine glands and kidney function. Some of these are common, such as thyroid gland disorders. Thyroid hormones directly affect kidney development, growth, glomerular filtration rate, renal transport, and electrolyte homeostasis. On the contrary, the kidney is involved in thyroid gland functioning—patients with chronic kidney disease (CKD) and a decreased filtration rate are at higher risk for thyroid dysfunction via iodine retention, metabolic acidosis, selenium deficiency or hormone depletion in NS, or peritoneal dialysis [9]. This review aims to systematically analyze the correlation of all endocrine gland disorders with NS and to emphasize possible bidirectional correlation (Table 1).

## 2. Pineal Gland

The pineal gland synthesizes serotonin, N,N-dimethyltryptamine, and melatonin—a serotonin-derived hormone that regulates sleep patterns in accordance with diurnal cycles [10]. During the dark phase, serotonin levels decrease as it is converted to melatonin, which is then secreted directly into the bloodstream and cerebrospinal fluid [11]. Due to its highly lipophilic nature, melatonin reaches virtually all cells in the body, binding to MT1 and MT2 receptors that are present in various tissues, including the kidneys [11].

While the physiologic functions of melatonin associated with sleep regulation, reproduction, and thermoregulation, as well as its role in the pathophysiology of neuropsychiatric diseases, have been known for a long time, its implication in kidney diseases has been studied recently. Melatonin receptors are predominantly expressed in the proximal tubule, and the impairment of endogenous circadian melatonin secretion was noticed in CKD [11]. Melatonin has potent anti-inflammatory and antioxidant effects via direct neutralization of the reactive species, induction of endogenous antioxidant enzymes, neutralization of the free radicals, inhibition of nitric oxide synthase, and inhibition of principal pro-inflammatory or apoptotic cytokines TNF-α and NF-κB [12,13]. Experimental studies on rat models have shown that treatment with melatonin can regulate sympathetic activity and decrease levels of profibrotic markers, such as collagen, alpha-smooth-muscle actin, and transforming growth factor-β. Deterioration of kidney function measured by the increases in plasma creatinine and proteinuria as well as pathological findings of chronicity, i.e., glomerulosclerosis, tubular atrophy, and interstitial fibrosis, was significantly ameliorated after treatment with melatonin [13]. Besides the above-noted, melatonin has beneficial effects on cell proliferation by enhancing the apoptosis of malignant cells and intraglomerular blood pressure regulation [11]. In humans, nocturnal electrolyte excretion is only 50% of that during the light period [14,15]. Dysregulated melatonin secretion at night was associated with renin–angiotensin–aldosterone activation. Therefore, bedtime consumption of melatonin can ameliorate the non-dipping type of hypertension [16]. 

Membranous nephropathy (MN) in adults is one of the most common glomerular diseases that manifest with NS. Several experimental studies on rat models have analyzed the possible therapeutic effect of melatonin in NS. Rats treated with melatonin had lower proteinuria and a significant attenuation of glomerular injury, with fewer immunocomplex deposits [17]. Further analysis showed that the CD19(+) B-cell population was decreased and interleukin-10 (IL-10), as an anti-inflammatory cytokine, was increased in those rats [17]. Melatonin also upregulated heme oxygenase 1 (HO1), an oxidative-stress-induced oxygenase, and its inhibition via SnPP showed kidney protection in rats with NS [17]. Huang et al. provided further knowledge by studying the role of melatonin receptors in the pathophysiology of MN in experimental models. In NS, melatonin receptor 1A (MTNR1A) expression was significantly decreased in renal tubular epithelial cells [18]. Molecular studies showed that the transcription factor pituitary homeobox-1 (PITX1) promoted its expression by direct binding to its promoter, and PITX1 kidney expression was decreased in NS. Authors have concluded that downregulated PITX1 leads to decreased levels of MTNR1A in tubular epithelial cells, which increases the future risk of MN [18]. Melatonin relieved the endoplasmic reticulum stress response in rats by decreasing glucose-regulated protein 78, phosphoinositol-requiring enzyme1α, and ATF6 levels, as well as suppressed the pro-apoptotic IRE1α/JNK signaling pathway [19].

## 3. Hypothalamus and Hypophysis

One of the primary functions of the hypothalamus is to serve as a critical link between the nervous and endocrine systems via the pituitary gland. The hypothalamus synthesizes oxytocin and vasopressin, which are subsequently released into the bloodstream through the posterior pituitary. Additionally, the hypothalamus contains parvocellular neurosecretory cells that secrete corticotropin-releasing hormone (CRH), growth-hormone-releasing hormone (GHRH), gonadotropin-releasing hormone (GnRH), thyrotropin-releasing hormone (TRH), and dopamine, which act on the anterior pituitary. The pituitary gland, also referred to as the hypophysis, secretes several key hormones, including adrenocorticotropic hormone (ACTH), growth hormone (GH), follicle-stimulating hormone (FSH), luteinizing hormone (LH), thyroid-stimulating hormone (TSH), and prolactin (PRL; Figure 1). Collectively, these hormones regulate numerous physiological processes, such as growth, blood pressure, energy metabolism, sexual function, thyroid activity, pregnancy, childbirth, lactation, renal water and salt balance, thermoregulation, and analgesia.

### 3.1. Adrenocorticotropic Hormone

By the late 1960s, ACTH had been replaced by synthetic oral glucocorticoids due to the ease of their administration and the belief that ACTH acted by stimulating the production of corticosteroids. In recent years, the use of ACTH therapy for the treatment of proteinuria due to NS has been heavily explored since nearly 20% of patients with NS are resistant to glucocorticoids, the golden standard for NS treatment, and 50% of glucocorticoid-sensitive patients experience frequent relapses [20]. Adrenocorticotropic hormone (ACTH) acts as an antagonist of the melanocortin system by binding to all five melanocortin receptors expressed in podocytes, glomerular cells, and various immune cells, playing a significant role in anti-inflammation, lipolysis, and modulation of exocrine function [21]. ACTH binding directly to podocytes stabilizes synaptopodin, reduces foot process effacement and apoptosis, improves histological signs of renal injury, decreases glomerular permeability, and consequently, reduces proteinuria [20]. Thus, ACTH therapy has demonstrated antiproteinuric, lipid-lowering, anti-inflammatory, and renoprotective effects [22]. Additionally, ACTH aids in the clearance of anti-PLA2R antibodies expressed in glomerular podocytes, which cause inflammation and death of surrounding renal tissue, leading to increased proteinuria [21]. Currently, anti-PLA2R antibodies serve as markers of the immunological activity of MN. Clinically, a reduction in proteinuria was observed after ACTH treatment in 70% of MN patients and 42% of FSGS patients [21]. Although further research is necessary, ACTH appears to be a promising agent for reducing proteinuria and disease flares in conditions such as IgA nephropathy, systemic lupus erythematosus, and minimal change disease. However, we are still lacking randomized studies that directly compare the safety and effectiveness of ACTH with that of oral glucocorticoids. Disadvantages of ACTH therapy include the need to be injected and the high cost.

### 3.2. Growth Hormone

Growth hormone (GH) is a 22 kDa protein secreted by the anterior pituitary gland in a pulsatile and predominantly nocturnal pattern, particularly during puberty, playing a vital role in postnatal growth and various biological functions, including metabolism and homeostasis [23]. GH and its mediator, insulin-like growth factor-1 (IGF-1), exert numerous effects on the kidneys, where they act synergistically. GH and IGF receptors are abundantly expressed in glomerular (mesangial and podocyte) and tubular cells. These receptors regulate glomerular hemodynamics, renal gluconeogenesis, tubular electrolyte reabsorption, renal activation of vitamin D, and Klotho expression [23]. In nephrotic syndrome, the levels of IGF-1 decrease, primarily due to the glomerular filtration of IGF-1-containing binding protein complexes [24].

The impact of glucocorticoids on bone is specifically relevant in children with nephrotic syndrome exposed to a long course of treatment, and it is proportional to the dose [25]. They cause a decrease in bone formation by osteoblastic inhibition in trabecular bone and might interfere with growth at several levels, including GH release, the secretion and action of IGF-1, and the inhibitory effect of the chondrocyte [26]. They also cause bone maturation delay, hypogonadism, pubertal delay, and IGF-1 inhibition [27]. A major cause of growth retardation in patients with NS in the pre-corticosteroid era was malnutrition, secondary to poor appetite, urinary protein loss, and malabsorption due to bowel edema [25]. Loke et al. for the first time conducted a study on eight children with steroid-dependent NS to whom GH was administered [26]. Their results have shown that one year of GH therapy had a significant impact in improving the height standard deviation score, height velocity, bone mineral density, and lean body mass of children, without significant adverse effects. However, there are only several randomized controlled trials with a limited number of patients, so a general conclusion cannot be drawn. The loss of IGF-1 and IGFBP-3 (predominant IGF-binding protein) in NS causes growth defects as well, whereas glucocorticoids can cause elevation of serum IGF-1 levels, suggesting potential development of IGF resistance [25]. 

In experimental studies involving patients with acromegaly, excess GH has been shown to adversely affect kidney function. This includes inducing glomerular hyperfiltration, renal hypertrophy, thickening of the glomerular basement membrane, proliferation of the mesangial matrix, and ultimately leading to conditions such as glomerulosclerosis or podocyte injury and detachment. The aforementioned effects are particularly pronounced in patients with uncontrolled type 1 diabetes mellitus accompanied by persistent albuminuria [23]. Conversely, GH deficiency is associated with a decrease in the glomerular filtration rate. GH induces Notch1 signaling in podocytes, a pathway that generally participates in the development and homeostasis of various organs and contributes to proteinuria in diabetic nephropathy. Inhibition of GH-induced Notch1 signaling may be a promising strategy for preventing diabetic nephropathy, a common cause of NS in the adult population. Additionally, GH has demonstrated nephroprotective effects in cisplatin-induced nephropathy by mitigating oxidative stress and silencing inflammation [23]. In pediatric patients with NS, IGF-1 levels were found to be decreased, and the proximal tubular filtrate activated IGF-1R, suggesting its bioactivity [23]. Furthermore, the tubular filtrate in NS stimulated the synthesis of collagen types I and IV, indicating that the excessive loss of IGF-1 in the nephrotic state may promote tubulointerstitial fibrosis and progressive kidney disease [28].

### 3.3. Prolactin

Elevated prolactin levels in chronic kidney disease (CKD) primarily result from reduced clearance and heightened secretion [29]. The prevalence of hyperprolactinemia among CKD patients varies widely, ranging from 30% in early stages to as high as 65% in those undergoing hemodialysis [30]. Regulation of prolactin secretion involves inhibition by hypothalamic dopamine [31]. Clinically, hyperprolactinemia typically presents with symptoms such as amenorrhea, galactorrhea, and hypogonadism [29]. NS can be associated with some cases of hyperfunctioning pituitary adenomas, but the association of proteinuria and hyperprolactinemia is an exceptional finding. Heras et al. described for the first time a case of a 40-year-old patient with asymptomatic nephrotic proteinuria (3.87 g/dU) and macroprolactinoma, with a favorable response to dopamine agonist therapy with NS remission [31]. The absence of hypoalbuminemia and other typical biochemical alterations of NS and edema could be explained by urine loss of a protein other than albumin. Proteinuria normalization with cabergoline implicates the possibility that the peak corresponded to urinary prolactin. The exact mechanism is unknown, but the authors proposed several pathophysiological mechanisms: protein hyperproduction or glomerular and tubular alterations, including tumor-related protein over-filtration. Namely, normal pathologic analysis speaks against the immune-mediated process. Furthermore, prolactin’s role was also speculated in pregnancies complicated with preeclampsia since it is elevated in human pregnancy in amniotic fluid. Prolactin can cause fluid retention, elevate arterial pressure, and potentiate responses to pressure agents [32].

### 3.4. Oxytocin

Oxytocin is an amino acid peptide hormone that has structural similarity to vasopressin. It has an antidiuretic effect and increases the urinary excretion of AQP2 (aquaporin-2), but this effect is not well elucidated [33]. However, in animals, the influence of oxytocin on renal function depends upon the species used, the dosage of the hormone, the degree of hydration, and the metabolic status of the animal [33]. In a rat model, oxytocin receptors were identified in the *macula densa* cell [34]. In experimental rat models with diabetes insipidus, acute administration of physiological doses of synthetic oxytocin caused a modest increase in the glomerular filtration rate, and chronic administration increased the glomerular filtration rate and plasma filtration by 40%. Nonetheless, oxytocin’s role in NS still needs to be elucidated.

### 3.5. Vasopressin

In patients with renal impairment, vasopressin levels are increased as a result of decreased clearance and impaired signaling in renal tubules [29]. Multiple authors have reported elevated AVP levels in NS [2]. A vasopressin-receptor antagonist interferes with action at the vasopressin receptor and induces effective aquaresis, unlike all other diuretics, which increase sodium excretion [2]. In nephrotic patients, the reduced blood volume stimulates AVP secretion, and an inverse correlation between blood volume and plasma AVP levels was observed [35]. In ten children with steroid-resistant NS, the combination of intravenous furosemide and oral tolvaptan (V2R antagonist that blocks vasopressin signaling) increased urine output [36]. Further studies are needed before clinical incorporation. To verify the correlation between AVP and water retention, Usberti and colleagues studied sixteen NS patients [35]. Plasma AVP decreased significantly in the first hour following water load only in control subjects, and a significant direct correlation was observed between plasma AVP and plasma osmolality in control subjects, but not in nephrotic patients. The results demonstrated a sustained volume-mediated secretion of AVP in the NS, which was responsible for the deterioration of water excretion [35]. Furthermore, the NS is also associated with “vasopressin escape”, characterized by low AQP2 expression in the collecting duct despite high vasopressin secretion [37]. This phenomenon was tested in 47 patients with NS. Their results have shown increased AVP secretion in NS compared with other invested groups. In patients with NS and a partial remission of NS combined, there was more than a ten-fold decrease in the median urinary AQP2 excretion compared with the control group. Finally, there was a negative correlation between the urinary AQP2 excretion and daily proteinuria [37]. However, in healthy rats and humans, vasopressin induced a marked increase in urinary albumin excretion via increased glomerular filtration and functional vasopressin V2 receptors, and was partly mediated by RAAS [38].

## 4. Pancreas

Acute pancreatitis is the inflammation of the pancreas with an increasing incidence of approximately 30 per 100,000 cases each year, with 5–30% mortality depending on severity [39]. Acute inflammation of the pancreas is a rare complication of NS, and the cause is likely related to the primary disease and medications [40]. Hao et al. reported atypical clinical presentation of acute pancreatitis in children with NS with often normal amylase and lipase levels, possibly due to the amylase-inhibiting factor released in hyperlipidemia, which is common in NS and inhibits the activity of amylase [40]. On the other side, proteinuria is a characteristic feature of severe acute pancreatitis that is not yet fully understood [41]. Many urine proteins may reflect aspects of the general inflammatory process that is occurring in early severe pancreatitis [41]. A possible pathophysiological explanation consists of the fact that edema and disorders of microcirculation in NS contribute to the slowing and reduction of blood flow in the pancreas, which leads to the development of hypoxia and metabolic disorders [42]. The autopsies of the pancreas in patients who died of glomerulonephritis revealed pathological changes in 35.7% of cases in the form of acute interstitial pancreatitis and necrosis [42]. 

On the other hand, NS is a common complication of unregulated diabetes mellitus and consequential diabetic nephropathy. Other glomerular diseases, superimposed on or unrelated to diabetic nephropathy, are common in 30% of patients with type 2 diabetes mellitus [43]. To emphasize this connection, Dogra et al. showed that patients with NS have an increased cardiovascular risk [44]. This is associated with endothelial dysfunction, lower plasma non-esterified fatty acids via impaired synthesis and release of nitric oxide, insulin resistance, glucose intolerance, and inflammation. Fasting insulin, fasting glucose, and the HOMA index are significantly higher in patients with NS, in comparison to healthy controls [44]. Jin et al. showed that in children with NS, fasting serum C-peptide levels were increased in the NS, suggesting that fasting serum C-peptide may be a protective factor [45]. Although, glucose intolerance and insulin resistance with compensatory enhanced beta-cell secretion, proteinuria, and glucocorticoid therapy can independently affect insulin sensitivity [46]. In conclusion, proteinuria is an independent risk factor for decreased insulin sensitivity [46]. Higher fasting and postprandial glucagon levels are associated with decreased glomerular filtration, increased albuminuria, and a risk of diabetic kidney disease in patients with type 2 diabetes mellitus [47].

## 5. Ovaries and Testicles

The kidney is one of the most estrogen-responsive (not reproductive) organs in humans. Estradiol modifies fluid and electrolyte homeostasis. Testosterone affects intrarenal hemodynamics, and animal data suggest that podocytes are a target for testosterone [48]. In NS, the hypothalamic–pituitary–gonadal axis is altered, and it has been proven that proteinuria negatively impacts gonadal function. Due to massive proteinuria, there is a loss of sex hormones and their binding proteins [48]. However, gonadotropin’s response to luteinizing releasing hormone stimulation was not significantly different in NS compared to the control group, suggesting an intact hypothalamic–pituitary axis in NS [49]. This was confirmed in the rat experimental model of NS. Female nephrotic rats had a rapid loss of the estrous cycle, which placed them in diestrus. Their hormonal evaluation showed a gradual decrease in E2, LH, and P4 concentrations, while there were no significant changes in FSH or testosterone values [50]. Histological examination of ovarian rat tissue in NS showed a considerable increase in the number of atretic follicles [50]. Furthermore, male rats had reduced levels of serum testosterone, elevated FSH, and increased urinary testosterone excretion [48,49]. The other study on male rats showed lower plasma testosterone, androstenedione, estradiol, and estrone concentrations, as well as high levels of LH, supporting the idea that testosterone urinary loss leads to the increased basal secretion of luteinizing hormone, presumably as a result of increased luteinizing releasing hormone secretion [49]. However, a catabolic state per se can be associated with nutritional deficiency and contribute to hypogonadism [48]. 

On the other hand, endogenous estrogens in animal models have shown antifibrotic and anti-apoptotic effects in the kidney. For instance, 17β-estradiol administration in experimental models after ovariectomy ameliorated glomerulosclerosis and tubulointerstitial fibrosis by protecting podocytes via upregulation of estrogen receptor β [22]. In human premenopausal women, oral contraceptive use is associated with macroalbuminuria, while postmenopausal women on hormone replacement therapy have a lower risk of albuminuria [22]. Still, this correlation needs to be evaluated in larger studies. NS in pregnancy presents a higher risk of both maternal and fetal adverse events, even in the absence of a decreased glomerular filtration rate or uncontrolled arterial hypertension [51]. The degree of proteinuria had an impact on both maternal and child outcomes, especially on the risk of preeclampsia, maternal CKD progression, maternal or fetal death, prematurity, small newborn for gestational age, or admission rate to the intensive care unit [52,53].

## 6. Thyroid Gland

Autoimmune thyroiditis impacts renal physiology through immunological and non-immunological mechanisms and, vice versa, kidney function can affect thyroid status [54]. Hypothyroidism is one of the most common endocrine disorders, with a prevalence of 4.6% of the population in the United States [55]. Thyroid hormone and its binding globulins are excessively excreted in urine in NS. Therefore, patients with NS may have subclinical or overt hypothyroidism and, additionally, corticosteroid therapy suppresses TSH secretion [56]. About 50% of patients with NS and a maintained glomerular filtration rate have low T4 due to urinary loss of T4-binding globulin and other thyroid-hormone-binding proteins (transthyretin and albumin) and the T4 bound to them, as well as intestinal edema and consequential impaired reabsorption [57,58]. Serum T3 concentrations may also be low due to decreased binding, and this is associated with abnormal platelet activation and an increase in platelet aggregation [59]. The euthyroid state is expected at the early stages of NS [60]. However, in prolonged and severe proteinuria, especially with concomitant low thyroid reserve, subclinical or overt hypothyroidism occurs, but the opposite is also possible with cases of reversible proteinuria and biopsy-proven glomerulonephritis, including MN, minimal change disease, or amyloidosis in association with hypothyroidism [60]. The risk of hypothyroidism is directly related to the severity of proteinuria regardless of sex and antibodies, and it might even predict a severe clinical manifestation and a poor clinical outcome of NS [61,62,63]. Iwazu et al. analyzed the relationship between thyroid function with renal hemodynamics and cholesterol metabolism in patients with NS [64]. They showed a positive correlation of FT3 with filtration fraction and an inverse correlation of FT4 with total cholesterol [64].

There is also an inverse relationship where severe hypothyroidism can mimic symptoms of NS, including reduced eGFR, proteinuria, dyslipidemia, and edema [55]. Furthermore, both subclinical and overt hypothyroidism are associated with greater odds of developing decreased eGFR and proteinuria (Figure 2) [55]. Hypothyroidism directly decreases renal sodium and water reabsorption, triggering the tubuloglomerular feedback, with preglomerular vasoconstriction and decreased glomerular filtration. It is also associated with a reduced beta-adrenergic response, decreased renin release and renin–angiotensin system, and reduced atrial natriuretic factor, all negatively impacting renal hemodynamics. Hyperthyroidism is more selective in the heart, with increased cardiac output, RAAS activation, and increased beta-adrenergic activation, and thus enhanced kidney filtration. 

Li et al. clinicopathologically studied 317 patients with NS [65]. Patients with hypothyroidism had higher proteinuria, creatinine, and lipid levels than those with normal thyroid function, but without differences in pathological types. However, after subdivision into five subgroups, MN was the most common pathologic type, both in the normal thyroid group and in the subclinical hypothyroidism group, while in the hypothyroid, low T3, and both low T3 and T4 groups, minimal change disease was the leading cause of NS [65]. Jain et al. analyzed the effect of proteinuria on thyroid function and its association with autoimmunity [66]. Patients were divided into two groups according to anti-TPO antibody positivity. The results showed that those with positive anti-TPO antibodies had more elevated TSH levels, proteinuria, and serum creatinine [66]. Interestingly, in a study involving 73 pediatric patients with nephrotic syndrome exhibiting thyroid function abnormalities treated with glucocorticoids, those who received thyroid replacement therapy demonstrated a shorter time to remission and higher serum albumin levels compared to those who did not receive thyroid replacement therapy.

While TSH is considered the most reliable and specific marker for primary hypothyroidism in the general population, this is not necessarily true in uremia because changes in TSH do not always reflect thyroid disease [22]. TSH is the last thyroid parameter that changes with progressive kidney dysfunction, and it is still more robust when facing non-thyroidal disease. When the glomerular filtration rate declines, thyroid hormone alterations increasingly occur, manifested as elevated TSH levels in 14% and low T3 levels in >75% of patients, causing non-thyroidal illness, where hormone alterations exist due to kidney disease other than alteration of the hypothalamic–pituitary–thyroidal axis [22]. Non-thyroidal illness is linked to increased cardiovascular mortality and morbidity. These effects are explained by several mechanisms: lower fT3 is associated with worse systolic left ventricular function, a low thyroid state exerts relative vasoconstriction, especially in the afferent artery, and is associated with a decrease in the circulating plasma volume, as well as linked to the development of atherosclerosis, impaired cardiac functioning, coronary artery calcification, arterial stiffness, endothelial dysfunction, inflammation, and malnutrition [22]. Medications that can suppress thyroid hormone metabolism include corticosteroids—the cornerstone therapy in NS—as well as amiodarone, propranolol, and lithium [22]. Testosterone deficiency has also been implicated in impaired deiodinase activity, although this connection is still unexplained. 

The correlation between calcitonin and NS is not clearly explained. Koopman et al. reported a unique case report of a patient with medullary thyroid carcinoma who developed NS caused by systemic calcitonin amyloidosis involving the kidney [67]. This type of localized amyloidosis occurred due to ACal fibrils, which were formed from the calcitonin hormone that is produced by the neoplastic C cells. Unlike other forms of amyloidosis, ACal in medullary thyroid carcinoma is not usually related to systemic amyloidosis, and the development of systemic ACal amyloidosis could be related to prolonged high circulating calcitonin levels [67].

## 7. Parathyroid Gland

Calcium disequilibrium and consequential bone turnover are common in patients with NS [68,69]. Low serum calcium and vitamin D levels are due to the loss of protein-bound calcium and the vitamin D major carrier, the D-binding protein, through urine [68,69]. Beyond its role in bone homeostasis, vitamin D has a plethora of different functions essential for good health, including renin–angiotensin–aldosterone system regulation, erythropoiesis regulation, anti-proliferation, immunomodulation, cardiovascular and overall mortality reduction, as well as the structural integrity of the slit diaphragm preservation [70]. Serum calcium in the combination of serum albumin, creatinine, and cholesterol improves the diagnostic sensitivity of steroid-sensitive versus steroid-resistant NS [70]. Thus, serum calcium may be used as an equivalent marker in the early diagnosis and treatment of NS [71]. The urinary loss of vitamin D in NS with preserved glomerular filtration might result in a decrease in blood levels of ionized calcium, secondary hyperparathyroidism, and enhanced bone resorption. As a consequence, osteomalacia, as evidenced by defective mineralization and increased osteoid volume, is observed [72]. It is important to emphasize that decreased vitamin D and the negative effects of long-term glucocorticoid therapy on mineral bone disease persist in the NS remission phase. The Z-scores of densitometry tests were positively correlated with vitamin D levels, and PTH levels were higher in patients with osteoporosis [73]. Furthermore, Banerjee et al. analyzed 48 patients with relapsed NS and 31 patients in NS remission, in comparison to controls [74]. Total vitamin D levels were the highest in the control group and the lowest in NS relapse and correlated negatively with proteinuria [74]. There was no difference in PTH levels between groups. When analyzing PTH levels in either glucocorticoid-treated or edematous patients of NS, no changes were observed, suggesting that the acute elevation in PTH after prednisone treatment was an acute phenomenon [75]. Children with relapsing NS (previously treated with corticosteroids) had higher levels of sclerostin and FGF-23, as markers of bone metabolism, which can indicate bone metabolism disorders. The significance of these observations requires further research [76]. Serum FGF23 levels begin to rise in NS patients before CKD G1, indicating that increased FGF23 levels are linked to both the progression of nephritis and early detection of abnormal mineral metabolism in patients with NS [77]. The co-activator for FGF23 binding to its receptors, Klotho, is a transmembrane protein also responsible for systemic mineral homeostasis maintenance by regulation of vitamin D and parathyroid hormone, as well as anti-inflammation, antifibrosis, and antioxidation [78]. A decreased Klotho level increased the purinergic receptor P2X7, causing cell apoptosis or necrosis, as well as suppressed the hyperglycemia-mediated glomerular endothelial injury and activation of the Wnt/β-catenin pathway in mice models of diabetic nephropathy, implicating Klotho’s possible role in pathophysiology of NS [78].

Finally, calcimimetics are extensively employed in managing secondary hyperparathyroidism in CKD by allosterically enhancing the calcium ion sensitivity of the calcium-sensing receptor (CaSR). Recently, cinacalcet has been postulated to play a novel role. Studies involving CaSR knockdown in podocytes and podocyte-specific CaSR knockout mice have suggested a stabilizing effect on the actin cytoskeleton and cell adhesion, resulting in reduced proteinuria [79]. Future investigations are warranted to delineate their potential therapeutic implications in NS.

## 8. Adrenal Gland

The over- and under-filling theories are two aspects of the same spectrum of NS. The “underfill hypothesis” suggested that massive proteinuria and hypoalbuminemia led to fluid extravasation into the interstitial space, causing intravascular hypovolemia and activation of neurohormonal compensatory mechanisms, which increased the retention of salt and water. Vasopressin, renin, aldosterone, and catecholamines are, therefore, stimulated in patients with NS via a reduction in the effective circulatory blood volume [80,81]. On the contrary, the “overfill theory” postulates that the edema is due to primary renal sodium retention, progressive extracellular fluid volume expansion, and edema formation via an overflow mechanism. Sodium retention is a major clinical feature of NS. However, it can happen in some patients in the absence of activation of the RAAS, suggesting an intrinsic defect in sodium kidney excretion, i.e., sodium transporters might be activated by factors present in nephrotic urine [82]. Beyond sodium reabsorption and blood pressure regulation, aldosterone is involved in kidney inflammation and fibrosis via activating the mineralocorticoid receptor in podocytes, mesangial cells, epithelial cells, and myeloid cells. Thus, treatment with aldosterone antagonists seems to be effective in reducing proteinuria and kidney fibrosis [82,83].

Although serum cortisol can be reduced in NS due to the loss of cortisol-binding globulin in the urine, serum-free cortisol concentrations are normal, and symptomatic hypocortisolism does not occur. On the contrary, data from the literature have shown that 84.6% of patients with autonomous cortisol hyperproduction have increased albuminuria unrelated to blood pressure or plasma glucose levels [84]. The increase in albuminuria in Cushing’s syndrome can be explained by abnormal lipid metabolism because GC increases visceral fat content, which leads to the increase in plasma-free fatty acid, finally binding to serum albumin and increasing the urinary albumin excretion or activating the protein kinase-C pathway in kidney endothelial cells [85]. ACTH or glucocorticoid administration causes an increased glomerular filtration rate, while chronic exposure can cause a decreased filtration rate, glomerular dysfunction, and proteinuria, with pathological evidence of glomerulosclerosis. Tubular dysfunction manifested as an impaired urinary concentrating ability and electrolyte disequilibrium [86]. However, only aldosterone, but not cortisol, can act within target organs selectively, because 11β-hydroxysteroid dehydrogenase type 2, whose protein was detected in human glomerular podocyte and decreased in diabetic rats, converts active cortisol to inactive cortisone [84]. Progressive kidney disease impacts cortisol metabolism and urinary clearance of cortisol metabolites [87]. Thus, decreased activity of 11β-HSD2 in diabetes mellitus does not render cortisol inactive, leading to a mineralocorticoid-like action due to cortisol in the kidney. Meuwese et al. emphasized that only a few studies investigated adrenal gland hormone levels in CKD/NS, with conflicting results, especially the problems presented by medications, such as RAAS inhibitors or glucocorticoids [22]. Dehydroepiandrosterone (DHEA) and dehydroepiandrosterone sulfate (DHEA-S) are secreted from the zona reticularis. Both low circulating concentrations of DHEA and DHEA-S correlate with the progression of glomerular injury in patients with diabetes mellitus, negatively correlating with proteinuria and blood creatinine [88,89]. A possible explanation lies in the effect of PPARα activators that can modulate immune function, inflammation, and oxidative stress [88]. Finally, catecholamines are also involved in modulating several aspects of renal physiology, including renal blood flow, glomerular filtration, tubular transport, and renin and erythropoietin release. Because increasing evidence suggests that renal ischemia resulting from various stress states plays a major role in the etiology of glomerulonephritis, the renal hemodynamic effect of continuous infusions of catecholamines was studied in a dog model. The primary response was a decrease in renal blood flow. As the rate of infusion was increased, a progressive decrease in the number of functioning glomeruli followed [90]. Szokol et al. analyzed the effect of adrenalectomy on proteinuria in a rat model [91]. The results showed that proteinuria was significantly diminished after adrenalectomy both in the hypertensive and control groups, and moderately non-selective glomerular proteinuria persisted in the spontaneously hypertensive rats, referring to a damaged glomerular barrier [91]. Further studies are needed to elucidate this correlation in humans.

## 9. Bidirectional Connection of Kidneys and Endocrine Glands—Current Knowledge and Future Directions

Endocrine disorders are very common in patients with NS and, conversely, NS itself, as well as its treatment options—especially glucocorticoids, which form the basis of the therapy, or other essential but aggressive medications—can exacerbate some hormonal disorders. The exact prevalence of these complications remains to be determined due to the complexity of the human endocrine system, which involves many glands. While cardiovascular, metabolic, hematological, and infectious complications are more prevalent, NS-associated endocrine disorders are often overlooked. In clinical practice, interpreting hormonal disturbances in NS is challenging, as it requires differentiating between pathological changes and compensatory responses or expected disease progression, which is crucial for optimal patient care [92].

The endocrine system, in its complexity, participates in the functioning of the entire organism. All endocrine glands, including the pineal gland, hypothalamus, pancreas, ovaries, testicles, thyroid gland, parathyroid glands, and adrenal glands, along with their hypothalamic–pituitary–adrenal axis, have reciprocal effects. The impact of endocrine glands on the body, including the kidneys, as well as the effect of kidney diseases, including NS, on the endocrine glands is extremely broad and diverse. This ranges from effects on the most basic molecules and receptors to effects on various electrolytes, directly influencing the patient’s clinical presentation. This already intricate network becomes even more complex with further deterioration of kidney function. However, although NS is commonly associated with preserved glomerular filtration rate (GFR), approximately 20–30% of patients experience a decrease in GFR, further influencing endocrine abnormalities [93]. Decreased clearance of active molecules and hormones due to reduced GFR complicates and sometimes renders it impossible to monitor endocrinological problems [92]. Literature reports indicate sex hormone deficiencies, low levels of IGF-1 and calcitriol, as well as increased prolactin, cortisol, and insulin concentrations, and thyroid hormone disturbances in these patients, all of which affect their quality of life [94]. The clinical consequences of these endocrine abnormalities are further compounded by CKD syndromes, such as the accumulation of uremic toxins, diminished clearance of molecules, acid-base or electrolyte disturbances, abnormal glycosylation, CKD treatment options, malnutrition, impaired adaptive responses, reduced response to stimulating hormones, and inflammation [95,96].

Moving forward, advancing our understanding of the bidirectional relationship between the kidneys and endocrine glands presents several promising avenues for research and clinical practice. Firstly, leveraging advancements in molecular biology and genetics holds potential for uncovering the intricate mechanisms underlying endocrine disturbances in kidney diseases. By identifying genetic predispositions and elucidating molecular pathways, we can pave the way for targeted therapeutic interventions tailored to individual patient profiles. Secondly, the development of novel biomarkers is critical to accurately assess endocrine function in the context of renal disorders. These biomarkers would facilitate early detection, precise monitoring, and timely intervention in patients at risk of developing endocrine abnormalities. Thirdly, integrated therapeutic approaches that consider the reciprocal impacts of kidney and endocrine dysfunctions are essential. Customized treatment protocols based on comprehensive assessments of both renal and endocrine health can optimize patient outcomes and enhance quality of life. Longitudinal studies are also paramount to delineating the progression of endocrine disorders in CKD and NS, providing insights into the natural history of these conditions and evaluating the efficacy of interventions over time. Furthermore, fostering interdisciplinary collaboration among nephrologists, endocrinologists, and primary care physicians is crucial for implementing holistic care models that address the complex interplay between renal and endocrine systems. By embracing technological innovations, such as artificial intelligence and machine learning, we can harness predictive analytics to anticipate and manage endocrine complications in renal disease proactively. Educating and training healthcare providers on the interconnected nature of renal and endocrine health will empower them to deliver informed, integrated care and adhere to updated clinical guidelines. Lastly, engaging patients in research endeavors to understand their perspectives and experiences with concurrent renal and endocrine disorders will guide the development of patient-centered approaches that prioritize individualized care and improve treatment outcomes. Together, these initiatives promise to advance our ability to manage and mitigate the multifaceted challenges posed by the bidirectional relationship between the kidneys and endocrine glands.

## 10. Conclusions

The significance of the topics addressed in this paper lies in their potential to illuminate the pathophysiology of fundamental organismal disorders and their intricate interactions. Understanding the bidirectional connections between NS and endocrine disorders is crucial for anticipating future implications and devising effective treatment strategies. Interventions aimed at managing one condition often impact the other, sometimes positively and at other times detrimentally. Therefore, addressing NS and associated endocrine disorders necessitates consideration of the varied bidirectional consequences that may arise. While our review outlined potential bidirectional connections, substantial gaps remain in understanding that require validation in clinical settings. We underscored the importance of adopting a comprehensive approach to managing every NS patient, emphasizing meticulous hormonal assessment, particularly in symptomatic individuals. Early and tailored interventions have the potential to mitigate morbidity and mortality in NS/CKD patients, underscoring the importance of prompt individualized care.

## Figures and Tables

**Figure 1 biomedicines-12-01860-f001:**
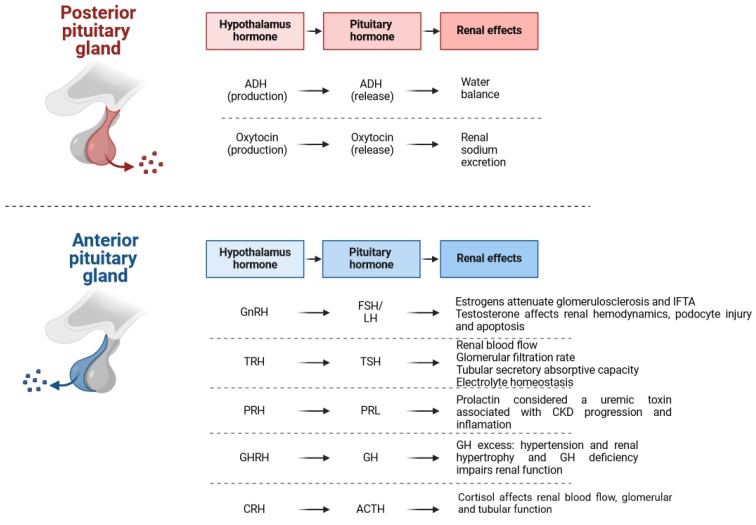
A review of pituitary gland hormones’ effects on kidneys. Created in BioRender.com. Abbreviations: adrenocorticotropic hormone (ACTH), growth hormone (GH), follicle-stimulating hormone (FSH), luteinizing hormone (LH), thyroid-stimulating hormone (TSH), prolactin (PRL), antidiuretic hormone (ADH), thyrotropin-releasing hormone (TRH), gonadotropin-releasing hormone (GnRH), growth-hormone-releasing hormone (GnRH), corticotropin-releasing hormone (CRH), prolactin-releasing hormone (PRH), and interstitial fibrosis and tubular atrophy (IFTA).

**Figure 2 biomedicines-12-01860-f002:**
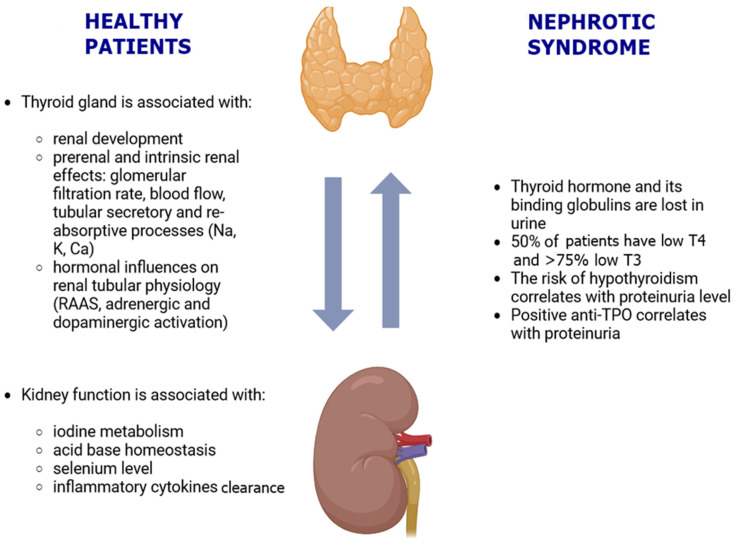
Bidirectional interplay of the thyroid gland and kidneys in healthy and NS patients. Abbreviations: Na, sodium; K, potassium; Ca, calcium; RAAS, renin–angiotensin–aldosterone system; TPO, thyroid peroxidase.

**Table 1 biomedicines-12-01860-t001:** A systematic review of endocrine glands, their hormones, and physiological roles.

Endocrine Gland	Hormones	Roles in Physiological Settings
Pineal gland (epiphysis)	-Serotonin -N, N-dimethyltryptamine -Melatonin	-Sleep patterns following diurnal cycles -Reproduction -Thermoregulation
Posterior pituitary gland	ADH	Water balance
	Oxytocin	Behavior, such as social binding, reproduction, and childbirth
Anterior pituitary gland	FSH/LH	-Trigger ovaries to estrogen production -Breast development -Menstruation/ovulation -Stimulate Sertoli cells to produce androgen-binding protein, thereby stimulating spermatogenesis
	TSH	Thyroid hormone regulation
	Prolactin	Breastfeeding and lactation
	Growth hormone	Postnatal growth, metabolism, and homeostasis
	ACTH	Cortisol homeostasis
Pancreas	Insulin Glucagon Somatostatin Pancreatic polypeptide	Glucose and metabolism homeostasis
Ovaries	-Estrogen -Androgen -Inhibin -Progesterone	-Menstrual cycle -Fertility
Testicles	Androgens (testosterone)	Fertility
Thyroid gland	-Triiodothyronine (T3) -Thyroxine (T4)	-Metabolic rate -Protein synthesis -Growth -Development
	Calcitonin	Calcium homeostasis
Parathyroid gland	Parathyroid hormone	Calcium homeostasis
Adrenal gland	Cortex: aldosterone, glucocorticoids, and androgens	-Electrolyte and water homeostasis -Stress response and glucose metabolism -Reproductive health and body development
	Medulla: catecholamines	Rapid reaction to stress situations

Abbreviations: adrenocorticotropic hormone (ACTH), follicle-stimulating hormone (FSH), luteinizing hormone (LH), thyroid-stimulating hormone (TSH), and antidiuretic hormone (ADH).

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
