# Peer review of "Endocrine Disorders in Nephrotic Syndrome—A Comprehensive Review"

_biomedicines, 2024, doi:10.3390/biomedicines12081860_

Round 1
Reviewer 1 Report
Comments and Suggestions for Authors
I have comprehensively reviewed the article titled ‘Endocrine Disorders in Nephrotic Syndrome – A Comprehensive Review’. In this article, the authors examine the relationship between nephrotic syndrome and endocrine disorders in bidirectional. The article provides a general overview of the literature and is a reference source on this subject. Overall, the paper is well written. I would suggest only a few minor changes to improve the quality of the article. I have marked these changes in the PDF file.
1. Classification of studies in the literature in terms of endocrine organs and presentation in tabular form
2. The paragraph on line 91 started with MN. It is not recommended to start paragraph introductions with abbreviations. Please write this clearly.
3. There is no explanation in the legends for IFTA in Figure 1. Please write an abbreviation.
4. Please add a reference for the information on line 294.
5. In Figure 2, it is written as pts instead of patients. Please write this clearly. Avoid non-standard abbreviations.
6. Parentheses were added by mistake on line 550. Please correct.
7. Some references are quite old. If possible, update the references.

Minor editing of English language required
Author Response
Dear Reviewer 1,
Dear Reviewer 1,
Please find attached our revised manuscript entitled “Endocrine Disorders in Nephrotic Syndrome – A Comprehensive Review” that we would like to resubmit for possible publication in the Biomedicines as a Review article after the peer-review process. We want to thank you for the time and effort invested in reading of our article. All of your respected objections were incorporated in the now, we hope, improved version of the manuscript. All changes have been highlighted in Word document with Text Highlight Colour with yellow colour and we are sending it in an attachment. In a further text we have written answers to your comment (Italic).
- Classification of studies in the literature in terms of endocrine organs and presentation in tabular form- Thank you for your comment. It is a little bit confusing what your idea was, so we hope we have added table properly.
- The paragraph on line 91 started with MN. It is not recommended to start paragraph introductions with abbreviations. Please write this clearly. – Thank you for your suggestion. We have changed that.
- There is no explanation in the legends for IFTA in Figure 1. Please write an abbreviation. - Thank you for your suggestion. We have added an explanation.
- Please add a reference for the information on line 294. – Thank you for your comment. The reference was added.
- In Figure 2, it is written as pts instead of patients. Please write this clearly. Avoid non-standard abbreviations.- Thank you for your suggestion. This abbreviation wa re-written as the whole word.
- Parentheses were added by mistake on line 550. Please correct. – Thank you for your comment. It was corrected.
- Some references are quite old. If possible, update the references. – Thank you for your comment. References were updated as possible. Some references are quite old since there is a lack of knowledge or obscure knowledge about some topics analyzed in the manuscript.
We hope that the reviewed version of manuscript is now suitable for publication in an esteemed journal as Biomedicines. For any further questions please e-mail me at maja.mizdrak@mefst.hr.
Yours sincerely,
Maja Mizdrak, MD, PhD

Reviewer 2 Report
Comments and Suggestions for Authors
Manuscript “Endocrine Disorders in Nephrotic Syndrome – A Comprehensive Review” is a review in the field of medicine dealing with kidney malfunction. Author review how disorders in various glands and organs including testicles impact the course of this disease.
Authors reviewed 94 references most of which are of 2020s including 2024. Presented references focused on the influence of hormones on the course of Nephrotic Syndrome. Manuscript includes the following paragraphs: Pineal gland, Hypothalamus and hypophysis, Pancreas, Ovaries and testicles, Thyroid gland, Parathyroid gland, Adrenal gland. Besides introduction and conclusions there is also “Bidirectional connection of kidneys and endocrine glands – current knowledge and future directions” section which give the insight on how information collected in the review can be used for the benefit of medical science.
The topics discussed in this manuscript are significant for shedding light on the pathophysiology of fundamental human disorders and their complex interactions. Understanding the bidirectional connections between neurological and endocrine disorders is essential for predicting future implications and developing effective treatment strategies. Thus presented manuscript could be published in the Biomedicines journal.
There are some minor issues that should be addressed before manuscript could be accepted for publication:
1. Manuscript need table of contents.
2. Sections 3 and 6 are supported with figures while others are not. I encourage you to add a table in the beginning of the manuscript describing all covered glands and organs, its hormones and corresponding effects. This will make the information presented easier to perceive.
Author Response
Dear Reviewer 2,
Please find attached our revised manuscript entitled “Endocrine Disorders in Nephrotic Syndrome – A Comprehensive Review” that we would like to resubmit for possible publication in the Biomedicines as a Review article after the peer-review process. We want to thank you for the time and effort invested in reading of our article. All of your respected objections were incorporated in the now, we hope, improved version of the manuscript. All changes have been highlighted in Word document with Text Highlight Colour with yellow colour and we are sending it in an attachment. In a further text we have written answers to your comment (Italic).
- Manuscript need table of contents. – Thank you for your suggestion. We have added a table of contents, although we are not sure whether this is appropriate in the manuscript as this one is.
- Sections 3 and 6 are supported with figures while others are not. I encourage you to add a table in the beginning of the manuscript describing all covered glands and organs, its hormones and corresponding effects. This will make the information presented easier to perceive. – Thank you for your suggestion. We have added a table, with all glands, their hormones and their roles in physiological settings, while their roles in nephrotic syndrome were explained in each section through the manuscript.
We hope that the reviewed version of manuscript is now suitable for publication in an esteemed journal as Biomedicines. For any further questions please e-mail me at maja.mizdrak@mefst.hr.
Yours sincerely,
Maja Mizdrak, MD, PhD
